# Prolonged Humid Heat Triggers Systemic Inflammation and Stress Signaling: Fluid Intake Modulates NF-κB, p38, JNK2, and STAT3α Pathways

**DOI:** 10.3390/ijms26115114

**Published:** 2025-05-26

**Authors:** Faming Wang, Caiping Lu, Ying Lei, Tze-Huan Lei

**Affiliations:** Centre for Molecular Biosciences and Non-Communicable Diseases, School of Safety Science and Engineering, Xi’an University of Science and Technology (XUST), Xi’an 710054, China; 22220226073@stu.xust.edu.cn (C.L.); 23120089006@stu.xust.edu.cn (Y.L.); thlei@xust.edu.cn (T.-H.L.)

**Keywords:** systemic inflammation, fluid intake, inflammatory signaling pathway, renal and hepatic stress

## Abstract

Prolonged exposure to extreme humid heat can induce systemic inflammation, organ stress, and hormonal imbalance. While fluid replacement is commonly recommended, its mechanistic efficacy under humid heat stress remains unclear. This study investigated the impact of fluid intake on thermoregulation, inflammation, organ function, and stress signaling during 8 h of humid heat exposure (ambient temperature: 40 °C, relative humidity: 55%) in 32 healthy young adults (20 males and 12 females). Participants completed two randomized trials: limited fluid intake (LFI, 125 mL/h) and full fluid intake (FFI, 375 mL/h). Core temperature (*T_core_*), inflammatory cytokines (IL-6, IL-1β, IFN-γ, TNF-α), organ stress markers (ALT, BUN), oxidative stress indices (MDA, SOD), and cortisol were assessed pre- and post-exposure. FFI significantly reduced post-exposure *T_core_* (37.8 ± 0.3 °C vs. 38.1 ± 0.3 °C, *p* = 0.046), mitigated cytokine elevations, and decreased BUN (blood urea nitrogen), ALT (alanine aminotransferase), and cortisol levels. Western blot analysis of PBMCs revealed that LFI activated NF-κB p65, JNK2, p38, and STAT3α phosphorylation, whereas FFI suppressed these responses. These findings demonstrate that adequate hydration attenuates heat-induced systemic and molecular stress responses. Our results highlight hydration as a key modulator of inflammatory signaling pathways during prolonged heat stress, offering insights into preventive strategies for populations vulnerable to climate-induced extreme heat events.

## 1. Introduction

Prolonged exposure to extreme humid heat can elicit a systemic inflammatory state, marked by elevated cytokine levels and immune activation [1,2]. In humans, acute thermal strain has been associated with increased circulating pro-inflammatory cytokines and physiological perturbations, including endothelial activation and organ-specific stress, particularly in the liver and kidneys [3]. While previous studies have largely emphasized the vulnerability of older adults to heat-related morbidity [4], the physiological response of healthy young adults under prolonged, uncompensable heat stress remains inadequately characterized.

Under conditions of sustained heat and humidity where evaporative cooling is impaired, thermoregulatory mechanisms may become insufficient [5]. This can result in a gradual but meaningful rise in core temperature (*T_core_*), which may trigger innate immune responses and stress signaling cascades [6,7,8]. In animal models, such responses involve the activation of canonical inflammatory pathways, including nuclear factor kappa B (NF-κB), mitogen-activated protein kinases (MAPKs), and the Janus kinase/signal transducer and activator of transcription (JAK/STAT) pathway [9,10,11,12,13]. These pathways regulate cytokine release such as interleukin-6 (IL-6) and interleukin-1β (IL-1β), which may contribute to systemic inflammation and stress in multiple organ systems [14,15]. However, the relevance and extent of these responses in healthy adults exposed to real-world thermal extremes, particularly over extended durations, are not well established.

Hydration status is a key physiological variable that influences cardiovascular stability, thermoregulation, and possibly the immune response during heat exposure [16,17,18]. While adequate fluid intake supports evaporative cooling and reduces circulatory strain [19,20,21], its potential to modulate inflammatory signaling remains unclear, especially under extreme humid heat where dehydration can accelerate. Many prior studies have used dry heat models or allowed unrestricted fluid intake, limiting insight into the specific role of hydration in the development of heat-associated systemic inflammation [22].

The present study investigates whether maintaining full fluid intake attenuates the physiological and inflammatory responses to prolonged humid heat exposure in healthy young adults. We evaluate cytokine levels and upstream signaling activity, focusing on the NF-κB, MAPK/JNK, and JAK-STAT pathways, in response to 8-h humid heat exposure with either limited or full fluid intake. Rather than implying clinical pathology, we use the term “*systemic inflammatory response*” to describe a pattern of biomarker changes resembling early inflammatory activation in the absence of infection or overt organ injury. We hypothesize that maintaining full fluid intake will limit the magnitude of cytokine signaling and reduce downstream stress responses during prolonged exposure to humid heat.

## 2. Results

### 2.1. Hydration Status

Urine specific gravity (USG) was significantly higher at the end of the exposure in the LFI condition compared to FFI (1.030 ± 0.004 vs. 1.020 ± 0.020, respectively, *p* < 0.01). The dehydration rate was markedly elevated in LFI (males: 3.0 ± 0.5%; females: 2.8 ± 0.4%), while no measurable dehydration occurred under FFI. Plasma electrolyte concentrations were also significantly higher in LFI than FFI, including sodium (146.2 ± 0.2 mmol/L vs. 141.1 ± 2.6 mmol/L), potassium (4.0 ± 0.2 mmol/L vs. 3.8 ± 0.2 mmol/L), and chloride (110 ± 11.5 mmol/L vs. 99.7 ± 2.4 mmol/L; all *p* < 0.01). These data confirm that LFI induced significant dehydration and electrolyte perturbation, which were prevented under FFI in both sexes (Table 1).

### 2.2. Body Temperatures

Core temperature (*T_core_*) was significantly higher in LFI than in FFI at the end of the exposure (condition × time interaction: *p* < 0.05). In contrast, mean skin temperature (*T_sk_*) did not differ between conditions at any time point in either sex (Table 1, *p* > 0.10).

### 2.3. Systemic Inflammation and Organ Function

Leukocyte and neutrophil counts were significantly elevated in LFI compared to FFI at the end of exposure, with condition × time interactions (*p* < 0.01; Table 1). Similarly, circulating interleukin-6 (IL-6) levels increased significantly under LFI compared to FFI (Figure 1A, *p* < 0.01), along with significant condition × time interactions. Additional pro-inflammatory cytokines, including IL-1β, IFN-γ, and TNF-α, also exhibited significant elevation in LFI relative to FFI (Figure 1B–D; all *p* < 0.01). Indicators of organ stress were elevated in LFI. Alanine aminotransferase (ALT) and blood urea nitrogen (BUN) levels were significantly higher in LFI than in FFI at the end of exposure, with interaction effects observed across conditions and time (Figure 1E,F; *p* < 0.01).

### 2.4. Oxidative Stress, Intestinal Cell Injury, and Cortisol Levels

Oxidative stress markers malondialdehyde (MDA) and superoxide dismutase (SOD) were significantly increased in LFI compared to FFI (Figure 1G,H; *p* < 0.01), with significant interaction effects (*p* < 0.01). Levels of intestinal fatty acid-binding protein (iFABP), a marker of enterocyte injury, and cortisol, a stress hormone, were also markedly elevated in LFI versus FFI (Figure 1I,J; all *p* < 0.01).

### 2.5. Inflammatory Signaling Pathways

Limited fluid intake (LFI) significantly increased the relative expression of phosphorylated NF-κB p65 (p-NF-κB p65), JNK2 (p-JNK2), p38 (p-p38), and STAT3α (p-STAT3α), each normalized to their respective total protein levels, in both males and females (*n* = 6 per sex group) compared to baseline (BL) conditions (all *p* < 0.05). In contrast, full fluid intake (FFI) significantly reduced the expression of these phosphorylated proteins compared to LFI (all *p* < 0.05). No significant differences were observed between BL and FFI for p-NF-κB p65 and p-JNK2 (all *p* > 0.05). The expression levels of phosphorylated JNK1 (p-JNK1) and STAT3β (p-STAT3β), normalized to their total forms, did not differ significantly across BL, LFI, and FFI conditions (all *p* > 0.05). Western blot analyses (Figure 2 and Appendix A) included corresponding total protein levels and GAPDH as a loading control to ensure accurate normalization.

## 3. Discussion

This study presents three key findings. First, prolonged exposure to extreme humid heat (8 h at 40 °C, 55% RH) triggered a systemic inflammatory response in healthy adults, reflected by elevated circulating cytokines and stress markers. Second, full fluid intake mitigated the magnitude of this response, reducing indicators of heat-induced physiological strain. Third, hydration status modulated the activation of canonical inflammatory signaling pathways, including NF-κB, MAPK/JNK, and JAK-STAT, with greater activation observed under limited fluid intake conditions.

The inflammatory response was evident from increased levels of IL-6, IL-1β, IFN-γ, leukocytes, and neutrophils in both sexes at the end of heat exposure, particularly under limited fluid intake (LFI). While previous studies using similar exposure durations have not consistently observed this level of inflammatory activation [4,23], the present study involved a greater elevation in core temperature (Δ*T_core_* ≈ 1.0–1.2 °C), likely contributing to the stronger cytokine response. These findings underscore the sensitivity of systemic inflammation to modest changes in core temperature and highlight the importance of thermal load when interpreting divergent outcomes across studies.

Notably, TNF-α levels remained unchanged, suggesting that TNF receptor-dependent signaling was not the primary upstream trigger for NF-κB activation in this context (Figure 1 and Figure 3). Instead, gut-derived mechanisms, such as endotoxin leakage through a compromised intestinal barrier, may have contributed [24]. Although circulating lipopolysaccharide (LPS) and Toll-like receptor 4 (TLR4) expression were not directly measured, elevated intestinal fatty acid-binding protein (iFABP), a marker of enterocyte injury, suggests intestinal disruption [25,26]. This aligns with prior work indicating that gut permeability and endotoxemia can activate TLR4 and initiate downstream inflammatory signaling during heat stress [27,28,29,30]. Future studies should include direct measurements of LPS and TLR4 to confirm this mechanistic link.

The rise in plasma cortisol further suggests activation of the hypothalamic–pituitary–adrenal (HPA) axis, indicating a physiological stress response to prolonged heat exposure. Cortisol may amplify cytokine signaling and contribute to systemic inflammation [30,31]. Among cytokines, IL-6 showed the most pronounced elevation and may have acted upstream of IL-1β and IFN-γ. Together, these factors likely contributed to the increases in blood urea nitrogen (BUN) and alanine aminotransferase (ALT), which serve as indirect indicators of renal and hepatic strain. However, since creatinine or cystatin C were not measured [32,33], we avoid concluding overt organ dysfunction and instead interpret the data as evidence of organ stress rather than clinical injury [34,35].

Importantly, fluid intake significantly influenced molecular signaling. Western blot analyses of peripheral blood mononuclear cells (PBMCs) showed that LFI increased the phosphorylation of NF-κB p65, JNK2, p38, and STAT3α, while full fluid intake (FFI) suppressed this activation. These findings suggest that hydration can downregulate inflammatory pathways that are typically activated in response to stress and cytokine signaling. In contrast, the phosphorylation of JNK1 and STAT3β remained unchanged, highlighting isoform-specific roles during heat exposure. JNK2 and STAT3α appear to be more responsive to thermal and osmotic stress, consistent with previous findings that these isoforms contribute more directly to cytokine production and immune activation [36,37,38]. These data provide mechanistic support for the protective role of hydration during extreme environmental stress.

Given that only a subset of participants (*n* = 6 per sex group) provided sufficient blood volume for PBMC isolation and downstream signaling analysis, caution is warranted in generalizing these mechanistic findings. The smaller subsample limits the strength of inference but still provides useful insight into intracellular responses under tightly controlled conditions. Future studies with larger biosample yields are needed to confirm the observed signaling patterns across broader populations.

### 3.1. Limitations

Several limitations warrant consideration. First, we did not directly measure TLR4 expression or circulating endotoxin (LPS), which limits our ability to confirm gut-derived inflammatory signaling. Second, BUN and ALT are indirect markers, and the lack of creatinine or cystatin C data precludes definitive assessment of kidney function. Third, the sample size for PBMC signaling analyses (*n* = 6 per sex group) was limited by the volume of blood required; not all participants consented to full blood sampling, and PBMC yield was insufficient in many cases. While this subgroup analysis provides preliminary mechanistic insight, larger samples are necessary for more robust conclusions.

Another limitation is that the rehydration fluid contained carbohydrates in addition to electrolytes. Although this reflects commonly consumed beverages in occupational and athletic contexts (e.g., sports drinks, oral rehydration solutions) [39,40], carbohydrates themselves can influence physiological responses—such as stimulating insulin secretion, enhancing intestinal sodium and water absorption, and modulating the hormonal stress response [41,42]. Therefore, the observed effects may not be solely attributable to fluid volume replacement. Future studies comparing carbohydrate–electrolyte solutions with plain water are necessary to isolate the specific contribution of fluid composition to systemic inflammation and cellular stress signaling during prolonged heat exposure.

Finally, a key limitation is the absence of a normothermic control group (i.e., participants exposed to thermoneutral conditions with and without fluid intake). While our design aimed to replicate realistic occupational or environmental heat stress scenarios, the lack of such a baseline group limits our ability to disentangle physiological and molecular effects attributable solely to humid heat from those modulated by fluid intake. Including a normothermic comparison group in future studies would provide important reference values and clarify the distinct contributions of environmental heat stress and hydration status to systemic inflammation and stress signaling pathways.

### 3.2. Physiological Relevance and Practical Significance

The observed cytokine elevations and signaling activation suggest that prolonged humid heat exposure with insufficient hydration induces a physiologically relevant systemic inflammatory state. However, we refrain from using clinical terms such as systemic inflammatory response syndrome (SIRS) given the absence of full diagnostic criteria (e.g., temperature, heart rate, respiratory rate, and white blood cell count thresholds) [43,44]. Rather, our findings indicate a systemic inflammatory response, one that reflects a state of heightened immune activation without necessarily indicating clinical pathology.

From a public health perspective, these results emphasize the importance of maintaining hydration during prolonged heat exposure. Adequate fluid intake suppresses cytokine activation and intracellular signaling that otherwise contribute to systemic stress and potential tissue injury. This has direct relevance for occupational, athletic, and environmental settings where prolonged heat exposure is common. Future research should explore individualized hydration strategies and real-time monitoring tools that could mitigate heat-related inflammation and reduce the risk of heat-associated health complications.

## 4. Materials and Methods

### 4.1. Ethical Approval and Participants

This study was approved by the Institutional Review Board of Xi’an University of Science and Technology (XUST-IRB224007). Thirty-two non-acclimatized, healthy young adults (12 females: 23.2 ± 1.9 years, body mass index: 22.2 ± 3.1 kg/m^2^; 20 males: 24.8 ± 2.3 years, body mass index: 23.9 ± 2.4 kg/m^2^) were recruited. All participants provided written and verbal informed consent prior to enrollment, in accordance with the Declaration of Helsinki.

### 4.2. Experimental Protocol

Each participant completed two 8-h extreme heat exposure trials (air temperature: 40 °C; relative humidity: 55%) under two hydration conditions: limited fluid intake (LFI) and full fluid intake (FFI). The order of trials was randomized and counterbalanced, with a 5-day washout period between sessions.

Prior to each trial, participants voided their bladders to ensure a urine specific gravity < 1.020, measured using an LH-Y12 meter (Lohand Biological, Hangzhou, China; accuracy: ±0.001), following recommendations by Oppliger et al. [45]. After a 15-min seated rest, baseline blood and urine samples were collected. A rectal thermistor was self-inserted (10 cm beyond the anal sphincter, YSI401, Yellow Springs, OH, USA), and monitoring instruments were attached.

Following a 10-min baseline period, participants entered a climate-controlled chamber (Espec Corporation, Osaka, Japan) for the 8-h heat exposure. During LFI, participants consumed 125 mL/h of an electrolyte beverage (lemonade with 0.4 mg zinc, 2.5 g carbohydrates, 25 mg sodium). During FFI, intake was 375 mL/h. The LFI volume reflects typical daily adult fluid intake in China and the U.S. [46,47]. Participants remained seated and engaged in light office tasks. A standardized 550 kcal Big Mac^®^ (McDonald’s, Des Plaines, IL, USA) was provided during the trial.

Rectal and skin temperatures and heart rate were recorded continuously. Blood pressure (BP) and perceptual responses were assessed every 60 and 30 min, respectively. Post-exposure measurements included body weight, urine, and blood samples.

Trials were immediately terminated if any of the following criteria were met: (1) rectal temperature reached 39.0 °C; (2) heart rate exceeded 85% of the participant’s age-predicted maximum (220 − age) [48]; (3) systolic blood pressure (SBP) fell below 90 mmHg or diastolic blood pressure (DBP) below 50 mmHg [49]; (4) the trial reached the full 8-h duration; or (5) the participant voluntarily withdrew due to severe discomfort, nausea, limb numbness, or difficulty breathing.

### 4.3. Measurements

#### 4.3.1. Blood Preparation

Venous blood samples were collected into serum tubes (additive-free vacuum blood collection tubes, KWS Medical Technology Co., Ltd., Shijiazhuang, China), allowed to clot, and centrifuged at 3000× *g* for 20 min at 4 °C. Sera were aliquoted and stored at −80 °C. EDTA-anticoagulated blood was centrifuged at 3000× *g* for 15 min at 4 °C to obtain plasma, which was also stored at −80 °C. Plasma was separated by centrifugation at 4 °C at 3000× *g* for 15 min, carefully extracted, and stored at −80 °C for further analysis. It is worth noting that peripheral blood mononuclear cell (PBMC) isolation requires a relatively large blood volume. As such, sufficient PBMCs were obtained from a subset of six participants (three males and three females) from the full cohort (*n* = 32) to enable downstream analyses. This subsampling approach was chosen to minimize participant burden while still enabling the mechanistic investigation of systemic inflammatory responses. Therefore, PBMCs were isolated from a subset of 12 participants (6 males, 6 females) to enable mechanistic analyses. This subsampling was chosen to balance scientific rigor with participant burden.

#### 4.3.2. Inflammatory Cytokines

Serum concentrations of IL-6 (EK106HS-02), IL-1β (EK101BHS-01), TNF-α (KE00367), and IFN-γ (EK180HS-AW1) were quantified using ELISA kits (Lianke Biotech, Hangzhou, China). Intra-assay coefficients of variation (CVs) were IL-6, 4.7%; IL-1β, 4.5%; TNF-α, 3.6%; IFN-γ, 4.3%. Detection limits were 0.02 pg/mL for IL-6 and IL-1β, 0.4 pg/mL for TNF-α, and 0.04 pg/mL for IFN-γ.

#### 4.3.3. Oxidative Stress, Intestinal Cell Injury, and Cortisol

Malondialdehyde (MDA) was measured using a kit from Nanjing Jiancheng Bioengineering Institute (CV: 2.3%, sensitivity: 0.5 nmol/mL), and superoxide dismutase (SOD) was measured using Servicebio (CV: 3%, sensitivity: 0.2 U/mL). Intestinal cell injury was evaluated by quantifying plasma intestinal fatty acid-binding protein (iFABP) via an ELISA kit (Youersheng, Wuhan, China). Plasma cortisol was measured with a kit from the same manufacturer.

#### 4.3.4. Organ Function Markers

Renal and hepatic stress were evaluated by serum blood urea nitrogen (BUN) and alanine aminotransferase (ALT), respectively. BUN was analyzed using an enzyme-labeled detector (Epoch, BioTeK Instruments Inc., Winooski, VT, USA) and ALT via an automatic biochemistry analyzer (Chemray 800, Radu Life Sciences, Shenzhen, China).

#### 4.3.5. Whole Blood Cell Counts

Capillary blood was analyzed for leukocyte count, hemoglobin, and hematocrit using an automatic hematology analyzer (Getein BHA-3000, Nanjing, China). All hematological values were corrected for plasma volume shifts using the Dill and Costill method [50].

### 4.4. Western Blotting

Peripheral blood mononuclear cells (PBMCs) were isolated from venous blood samples collected pre-exposure and 40–60 min post-exposure using Ficoll–Paque density gradient centrifugation. Following isolation, cells were washed in phosphate-buffered saline (PBS) and counted using a hemocytometer (SCC-M630, Servicebio, Wuhan, China). PBMCs were lysed on ice for 30 min in a lysis buffer containing PMSF (G2008-1ML, Servicebio), 50× Cocktail (G2006-250UL, Servicebio, Wuhan, China), a phosphorylated protease inhibitor (G2007-1ML, Servicebio), and Strong RIPA buffer (G2002-30ML, Servicebio). Lysates were centrifuged at 12,000× *g* for 15 min at 4 °C to remove cellular debris, and supernatants were collected. Protein concentrations were quantified using a bicinchoninic acid (BCA) protein assay kit (G2026-200T, Servicebio) according to the manufacturer’s instructions. Samples were normalized, mixed with protein loading buffer, vortexed briefly, and denatured at 95 °C for 10 min. Proteins were separated on 10% SDS-PAGE gels by electrophoresis at 150 V for 90 min and transferred onto polyvinylidene fluoride (PVDF) membranes at 350 V for 40 min. Membranes were washed with TBS-T (G0004-1L, Servicebio), blocked for 1 h at room temperature with a protein-free rapid sealing solution (G2052-500ML, Servicebio), and incubated overnight at 4°C with primary antibodies.

All blots were sequentially probed in the following order: (1) phosphorylated proteins (e.g., p-NF-κB p65, p-JNK1/2, p-p38, p-STAT3α/β), (2) total proteins (corresponding non-phosphorylated forms), and (3) GAPDH as a loading control. This probing order (phosphorylated → total → loading control) was selected to preserve phospho-epitopes, which are known to degrade upon membrane stripping. In cases where membrane stripping was necessary between antibody incubations, a mild stripping buffer (Servicebio) was used under validated conditions to ensure that protein integrity was maintained. However, to avoid potential issues with reprobing, parallel membranes were also used when necessary to allow independent probing of the phosphorylated and total protein forms. Both strategies (stripping vs. parallel membranes) were employed consistently across all blots, depending on the specific experimental requirements and the number of proteins being analyzed.

The primary antibodies used included Recombinant Anti-GAPDH (Mouse mAb, 1:4000, GB15002-100, Servicebio), NF-κB p65 (1:1000, cat. no. 8242, rabbit, Cell Signaling Technology, Danvers, MA, USA), phosphorylated NF-κB p65Ser536 (p-NF-κB p65, 1:1000, cat. no. 3033, rabbit, Cell Signaling Technology), p38 (1:1000, cat. no. 8690T, rabbit, Cell Signaling Technology), phosphorylated p38 MAPKThr180/Tyr182 (p-p38, 1:1000, cat. no. 4511, rabbit, Cell Signaling Technology), JNK (1:1000, cat. no. 67096, rabbit, Cell Signaling Technology), phosphorylated JNKThr183/Tyr185 (p-JNK, 1:1000, cat. no. 4668, rabbit, Cell Signaling Technology), phosphorylated STAT3Tyr705 (p-STAT3, 1:2000, cat. no. 9145, rabbit, Cell Signaling Technology), and STAT3 (1:1000, cat. no. 30835, rabbit, Cell Signaling Technology). After three washes with TBS-T, membranes were incubated for 40 min at room temperature on a horizontal shaker with HRP-conjugated secondary antibodies (goat anti-rabbit, 1:10,000, GB23303, Servicebio). Protein bands were visualized using ECL chemiluminescent detection reagents (G2014-100ML, Servicebio) and imaged using a chemiluminescence imager (SCG-W3000, Servicebio). Densitometric analysis of band intensity was performed using ImageJ software https://imagej.net/ij/ (accessed on 29 April 2025) (NIH, Bethesda, MD, USA). Results were expressed as the ratio of phosphorylated protein to total protein, with total protein normalized to GAPDH, serving as the loading control.

### 4.5. Statistical Analysis

Data were analyzed using SPSS (v20, IBM, New York, NY, USA) and GraphPad Prism (v7.00, GraphPad Software LLC, Boston, MA, USA). Levene’s test assessed the homogeneity of variance; the Kolmogorov–Smirnov test assessed the normality. If assumptions were violated, appropriate transformations or non-parametric methods were applied. To assess the effect of fluid intake on inflammatory and thermoregulatory responses over time, a two-way repeated measures ANOVA was conducted, with hydration condition (LFI vs. FFI) and time point as within-subject factors. Mauchly’s test was used to evaluate sphericity, and when violations occurred, Greenhouse–Geisser corrections were applied. When significant main or interaction effects were detected, post hoc pairwise comparisons were performed using paired t-tests with Bonferroni corrections to control for multiple comparisons. For Western blot analysis, Bonferroni-corrected pairwise comparisons were conducted to evaluate the effect of fluid intake on relative protein expression levels. Data are presented as means ± standard deviation (SD), with statistical significance set at *p* < 0.05.

## 5. Conclusions

This study demonstrates that hydration status plays a crucial role in modulating physiological responses to prolonged extreme humid heat exposure in both males and females. We observed that eight hours of continuous heat exposure at 40 °C with limited fluid intake resulted in elevated circulating cytokines (IL-6, IL-1β, IFN-γ); increased intestinal fatty acid-binding protein (iFABP), indicating potential intestinal cell injury; and heightened activation of inflammatory signaling pathways, including NF-κB, MAPK/JNK, and JAK-STAT. These molecular changes were accompanied by elevations in biomarkers associated with renal and hepatic stress (e.g., BUN and ALT). Importantly, these responses were attenuated by full fluid intake, which suppressed key signaling intermediates such as phosphorylated NF-κB p65 and JNK2.

Although our findings do not confirm clinical diagnoses such as systemic inflammatory response syndrome (SIRS), they reveal a pattern of physiological strain and subclinical inflammatory activation that could precede more severe outcomes in vulnerable populations. These results underscore the importance of maintaining adequate hydration during prolonged heat exposure as a strategy to limit excessive immune activation, preserve intestinal barrier function, and reduce organ stress. Future research should further explore individualized hydration strategies and their potential to mitigate the systemic consequences of environmental heat stress.

## Figures and Tables

**Figure 1 ijms-26-05114-f001:**
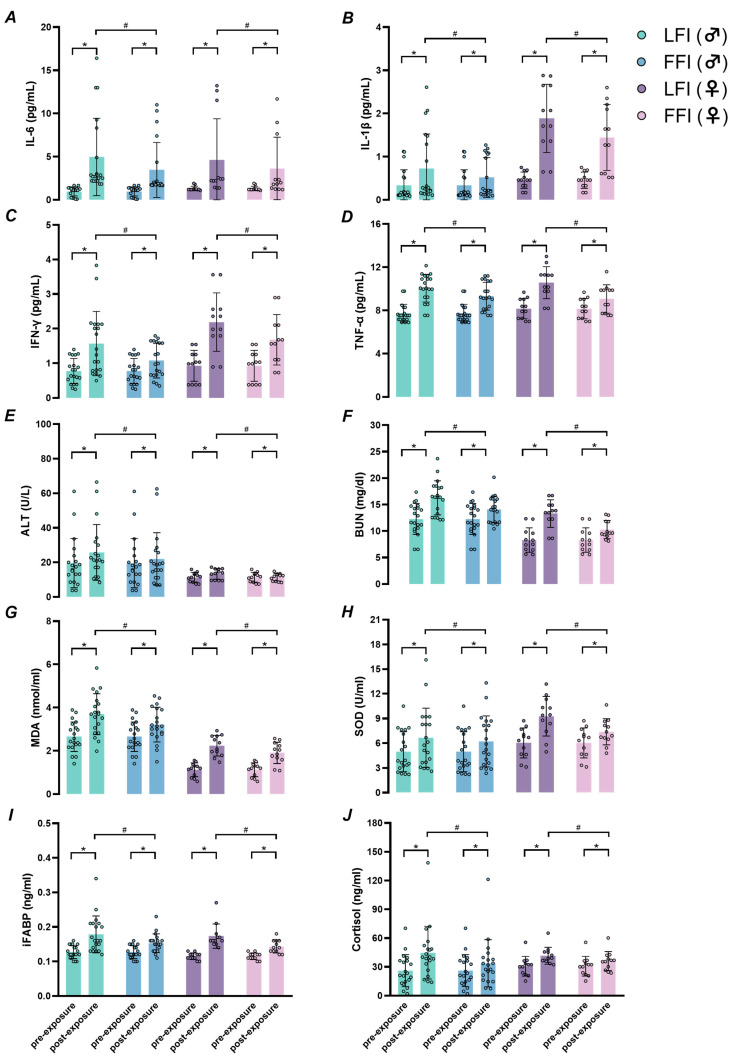
Systemic biomarkers measured before (pre-exposure) and after (post-exposure) the eight-hour prolonged humid heat exposure in the limited fluid intake (LFI) and full fluid intake (FFI) trials. (**A**), interleukin-6 (IL-6). (**B**), interleukin-1β (IL-1β). (**C**), interferon-γ (IFN-γ). (**D**), tumor necrosis factor-α (TNF-α). (**E**), alanine aminotransferase (ALT). (**F**), blood urea nitrogen (BUN). (**G**), malondialdehyde (MDA). (**H**), superoxide dismutase (SOD). (**I**), intestinal fatty acid-binding protein (iFABP). (**J**), cortisol. * Significant difference compared to pre-exposure (*p* < 0.05); # significant difference between FFI and LFI conditions (*p* < 0.05).

**Figure 2 ijms-26-05114-f002:**
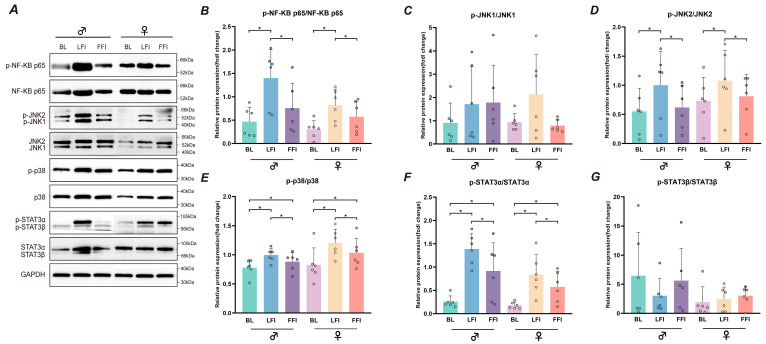
Scatter plots and representative Western blot analysis of protein expression in response to fluid intake during the eight-hour prolonged humid heat exposure. (**A**) Representative Western blot images of key proteins (p-NF-κB p65, total NF-κB p65, p-JNK1, total JNK1, p-JNK2, total JNK2, p-p38, total p38, p-STAT3α at ~86 kDa, total STAT3α at ~86 kDa, p-STAT3β at ~79 kDa, total STAT3β at ~79 kDa) and the loading control (GAPDH, used for normalization). (**B**–**G**) Relative expression levels of phosphorylated proteins, normalized to their respective total protein forms, in males (*n* = 6) and females (*n* = 6) under baseline (BL), limited fluid intake (LFI), and full fluid intake (FFI) conditions. (**B**) p-NF-κB p65/total NF-κB p65. (**C**) p-JNK1/total JNK1. (**D**), p-JNK2/total JNK2. (**E**) p-p38/total p38. (**F**) p-STAT3α/total STAT3α. (**G**) p-STAT3β/total STAT3β. * Significant difference (*p* < 0.05).

**Figure 3 ijms-26-05114-f003:**
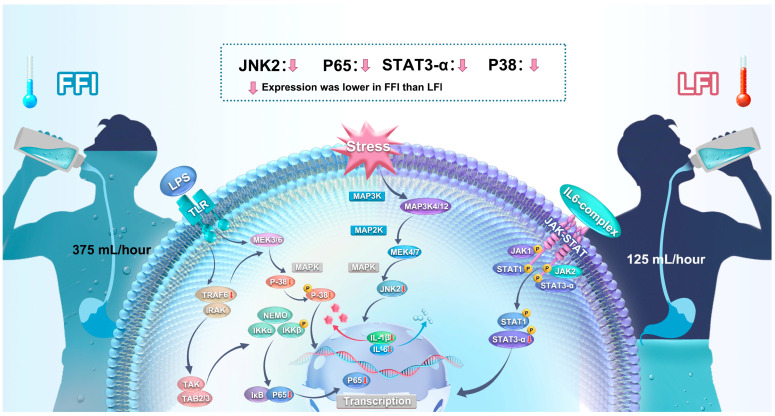
The role of fluid intake (FFI and LFI) in modulating inflammatory signaling pathways (NF-κB, MAPK/JNK, and JAK-STAT) during prolonged heat exposure. LFI, limited fluid intake; FFI, full fluid intake.

**Table 1 ijms-26-05114-t001:** Physiological biomarkers (including body temperatures, cardiovascular markers, blood biomarkers, and hydration status) and subjective perceptions (including thermal sensation, thermal comfort, skin wetness sensation, thirst sensation, and psychological stress) measured before (Pre) and after (Post) the eight-hour prolonged humid heat exposure in the limited fluid intake (LFI) and full fluid intake (FFI) trials. * Significant difference between post-exposure and corresponding pre-exposure values; # significant difference between FFI and LFI conditions. † Significant difference between males and females.

	LFI (Limited Fluid Intake)				FFI (Full Fluid Intake)			
	Males		Females		Males		Females	
Body temperatures	Pre	Post	Pre	Post	Pre	Post	Pre	Post
*T_core_* (°C)	36.9 ± 0.2	38.1 ± 0.3 *	37.0 ± 0.1	38.0 ± 0.3 *	36.9 ± 0.2	37.9 ± 0.4 *^#^	37.1 ± 0.1 ^†^	37.6 ± 0.2 *^#†^
*T_sk_* (°C)	33.8 ± 0.3	37.6 ± 0.4 *	33.0 ± 0.6	37.7 ± 0.4 *	33.9 ± 0.4	37.5 ± 0.5 *	33.5 ± 0.8 ^†^	37.4 ± 0.2 *
Cardiovascular	Pre	Post	Pre	Post	Pre	Post	Pre	Post
SBP (mmHg)	118 ± 7	112 ± 8 *	113 ± 10	105 ± 8 *	119 ± 9	111 ± 7 *	112 ± 5 ^†^	103 ± 5 *^†^
DBP (mmHg)	72 ± 6	63 ± 7 *	68 ± 8	58 ± 4 *	70 ± 5	63 ± 5 *	65 ± 5 ^†^	57 ± 5 *^†^
MAP (mmHg)	87 ± 6	79 ± 6 *	83 ± 7	74 ± 4 *	86 ± 6	79 ± 6 *	81 ± 4 ^†^	73 ± 5 *^†^
HR (bpm)	65 ± 8	118 ± 8 *	64 ± 7	112 ± 13 *	59 ± 5	110 ± 9 *	63 ± 4	113 ± 12 *
Blood biomarkers	Pre	Post	Pre	Post	Pre	Post	Pre	Post
Hematocrit (%)	43.4 ± 3.3	46.9 ± 2.0 *	30.8 ± 3.6	37.0 ± 3.7 *	43.9 ± 3.8	46.4 ± 2.0 *	34.1 ± 2.2 ^†^	37.0 ± 3.3 *^†^
Leukocytes (10^9^/L)	6.8 ± 1.2	13.6 ± 2.0 *	5.5 ± 0.7	10.8 ± 2.4 *	6.3 ± 1.2	12.5 ± 2.0 *	6.5 ± 0.7	10.2 ± 2.0 *^†^
Neutrophils (10^9^/L)	3.5 ± 0.7	8.7 ± 1.8 *	3.1 ± 0.6	6.6 ± 0.8 *	3.3 ± 0.7	8.1 ± 1.6 *	2.9 ± 0.6	5.7 ± 1.2 *^†^
K^+^ (mmol/L)	3.9 ± 0.1	4.0 ± 0.2 *	4.0 ± 0.2	4.1 ± 0.3 *	4.0 ± 0.2	3.8 ± 0.3 *	4.0 ± 0.2	3.8 ± 0.5 *
Na^+^ (mmol/L)	144.7 ± 3.4	146.1 ± 1.9 *	142.2 ± 5.3	142.7 ± 3.5 *	142.4 ± 3.0	141.1 ± 2.8 *	142.2 ± 5.3	130.0 ± 13.8 *^†^
Cl^−^ (mmol/L)	103.6 ± 3.6	110.1 ± 6.2 *	104.9 ± 2.2	105.4 ± 1.5 *	101.5 ± 3.9	99.5 ± 2.3 *	103.3 ± 4.6	97.1 ± 9.6 *^†^
PV change (%)	-	−6.7%	-	−7.4%	-	6.4%	-	8.7%
Hydration	Pre	Post	Pre	Post	Pre	Post	Pre	Post
USG (g/mL)	1.013 ± 0.005	1.030 ± 0.004 *	1.012 ± 0.003	1.030 ± 0.002 *	1.016 ± 0.007	1.014 ± 0.010 *^#^	1.011 ± 0.001	1.013 ± 0.003 *
Dehydration rate (%)	-	3.0 ± 0.5 *	-	2.8 ± 0.4 *	-	-	-	-
Sweat rate (g/h)	-	237 ± 48 *	-	165 ± 58 *	-	276 ± 80 *	-	214 ± 47 *^†^
Perceptions	Pre	Post	Pre	Post	Pre	Post	Pre	Post
Thermal sensation	0.8 ± 0.5	2.6 ± 0.4 *	0.7 ± 0.5	3.3 ± 0.5 *	0.8 ± 0.5	2.5 ± 0.8 *	0.7 ± 0.5	2.7 ± 0.5 *^†^
Thermal comfort	−0.5 ± 0.6	−2.5 ± 0.5 *	0.5 ± 0.8	−3.2 ± 0.9 *	−0.7 ± 0.6	−2.0 ± 0.7 *	0.3 ± 0.5 ^†^	−2.7 ± 0.5 *^†^
Wetness perception	−0.8 ± 0.6	−2.4 ± 0.5 *	−0.3 ± 0.5	−3.3 ± 0.5 *	−0.9 ± 0.6	−2.2 ± 0.5 *	−0.5 ± 0.5 ^†^	−2.7 ± 0.5 *^†^
Thirst sensation	2.1 ± 0.7	5.1 ± 0.8 *	1.2 ± 0.4	4.8 ± 0.8 *	1.6 ± 0.7	2.5 ± 1.0 *	1.8 ± 0.7	2.2 ± 0.4 *
Psychological stress	0.8 ± 0.7	6.5 ± 1.4 *	1.1 ± 1.0	6.3 ± 1.6 *	0.8 ± 0.8	5.0 ± 1.3 *	0.8 ± 0.9	5.5 ± 1.3 *

Note: -, not applicable. *T_core_,* rectal temperature. *T_sk_*, mean skin temperature. SBP, systolic blood pressure. DBP, diastolic blood pressure. MAP, mean arterial pressure. HR, heart rate. PV, plasma volume; USG, urine specific gravity.

## Data Availability

Data supporting the reported results can be found in Appendix A posted on figshare (https://doi.org/10.6084/m9.figshare.28899755.v2, accessed on 29 April 2025). Any further queries should be directly to the corresponding author of the article.

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
