# Peer review of "Prolonged Humid Heat Triggers Systemic Inflammation and Stress Signaling: Fluid Intake Modulates NF-κB, p38, JNK2, and STAT3α Pathways"

_ijms, 2025, doi:10.3390/ijms26115114_

Round 1
Reviewer 1 Report
Comments and Suggestions for Authors
Aim of this study is to examine if fluid intake can reduce elevated pro-inflammatory responses during prolonged (8 hours) humid heat exposure. Study included 32 young adults and two trials, one with limited fluid intake (LFI) and second with full fluid intake (FFI). FFI significantly reduced body temperature, production of pro-inflammatory cytokines (IL-1b, IL-6, IFN-g), blood urea nitrogen, ALT (alanine aminotransferase) and cortisol levels measured in blood compared to LFI. Also, analysis of PBMCs revealed that FFI suppressed NF-kB p65, JNK2, p38 and STAT3a which were elevated after humid heat exposure with LFI. These findings show how important is to stay hydrated during extreme heat events and how it can affect our immune system, providing directions for further research.
The Manuscript is well written, easy to follow, but some minor changes could be included:
- Introduction section can be enriched with additional references about immune system and what is known until now about effect of fluid intake on immune system.
2. Did you check if there is statistically significance between male and female participants?
Author Response
Please see attached file, thank you.

Reviewer 2 Report
Comments and Suggestions for Authors
The manuscript is interesting, well written and Authors presented results clearly with tables and figures. The aim of the study was to investigate the impact of fluid intake on thermoregulation, inflammation, organ function, and stress signaling during 8 hours of humid heat exposure in patients. The results demonstrate that adequate hydration decreased heat-induced systemic and molecular stress responses. Authors highlighted that hydration is a key modulator of inflammatory signaling pathways during prolonged heat stres.
Comments:
1.From physiology point view it is importent to follow the changes in organism induced by heat stress, not only to proof that hydration is important (what is obvious). I would suggest to change the title in such order-first disturbances induced by heat and after that the effect of fluid intake.
2.Hydration was with the fluid with carbohydrates what make difference comparing to pure water-carbohydrates by itself change some physiological processes.
3. There is lack of real control-patients staying in "normal" temperature, hydratation conditions , receiving or not that fluid.
4. I am recommending to add more newest publications.
In summary, the manuscript needs moderate changes.
Author Response
Please find attached the rebuttals, thank you.
